# Anti-TSH receptor antibodies (TRAb): Comparison of two third generation automated immunoassays broadly used in clinical laboratories and results interpretation

**José María López Ortega**[1]*, **Pilar Salvador Martínez**[1], **Delia Acevedo-León**[2], **Núria Estañ Capell**[2]

1 Autoimmunity and Allergy Division, Clinical Laboratories, Dr. Peset University Hospital, Valencia, Spain,
2 Hormones and Immunology Division, Clinical Laboratories, Dr. Peset University Hospital, Valencia, Spain

☯ These authors contributed equally to this work.
* lopez_josorta@gva.es

**Data Availability Statement:** Data cannot be shared publicly because we have no specific site to host it. Data are available from the Institutional

## Abstract

Graves' disease (GD) is the most common cause of hyperthyroidism in iodine-replete populations. It is an autoimmune disease caused by autoantibodies to the TSHR (TRAb). Although the diagnostic is mainly clinical, measuring TRAb improves accuracy and provides valuable prognostic information. The aim of this study was to compare the performance of two of the most widely used immunoassays i.e., EliA™ anti-TSH-R and Elecsys® anti-TSH-R. We have carried out a comparative study measuring TRAb by the two immunoassays in consecutive sera samples referred to the laboratory for TRAb measurement. Autoantibodies were measured in all samples in parallel by the two techniques. The two techniques were highly concordant as demonstrated by a Cohen's kappa of 0.82. At the manufacturer recommended cut-off, sensitivity of Elecsys® TRAb test was higher (100% vs. 96.6%), while specificity of the EliA™ TRAb test was higher (99.4% vs. 95.3%). In most patients TRAb are detected by any of two tests which are both well suited for Clinical Laboratories use. However, a higher specificity may constitute an advantage for measurement used not for screening but for diagnostic purposes, as anti-TSH-R is.

## Introduction

Graves' disease (GD) is an organ-specific autoimmune disease of the thyroid gland that affects predominantly women (ratio about 8:1) between 30 and 50 years old and is the most common cause of hyperthyroidism in iodine-replete populations [1]. The mechanism of hyperthyroidism in GD is the production of autoantibodies to the Thyroid Stimulating Hormone Receptor (TSH-R) that mimic the effects of the thyrotropin. The TSH-R belongs to the family of 7TM G-protein coupled receptors and is expressed by thyroid follicular cells and, to a much lower level, by thymocytes and fibroblasts of retro-orbital tissue [2]. Based on their functional effect on TSH-R, three types of antibodies can be considered: stimulating, blocking, and cleaving ("neutral" in biological activity terms) TRAb [3]. The stimulating TRAb are the most common and the cause of hyperthyroidism in GD patients [4].

Data Access / Ethics Committee (contact via e-mail: analisisclinicos_hpeset@gva.es) for researchers who meet the criteria for access to confidential data.

**Funding:** JML received the research grant CABAI1809 from Thermo Fisher Scientific. https://www.thermofisher.com/phadia/wo/es/our-solutions.html The funders had no role in study design, data collection and analysis, decision to publish, or preparation of the manuscript.

**Competing interests:** The authors have declared that no competing interests exist.

**Abbreviations:** AIT, autoimmune thyroiditis; AUC, area under the curve; ECLIA, electrochemiluminiscent immunoassay; FEIA, fluorescence enzyme immunoassay; GD, Graves' Disease; GD-R, GD in remission; GD-T, GD under treatment; LATS, long-acting thyroid stimulator; LoQ and UpQ, lower and upper limits of quantitation; LR, likelihood ratios; MISC, miscellaneous non-thyroid patients; MNG, multinodular goiter non-toxic; MoAb, monoclonal antibody; NDGD, newly diagnosed GD; OCO, optimum cut-off; OTD, other thyroid diseases; pTSHR, porcine TSH-R; ROC, receiver operating curve; TRAb, thyrotropin receptor autoantibodies; TSHR, thyroid stimulating hormone receptor.

The autoantibodies to the TSH-R are of high affinity but remain at a low absolute concentration as they are produced by a limited number of clones of B lymphocytes and plasma cells and this may explain why in a few patients, the response may shift from producing predominantly stimulating to blocking or neutral antibodies with the corresponding change in the state of the thyroid function and the clinical symptoms [5, 6].

Like other autoimmune diseases, GD is a chronic condition that requires early diagnosis to prevent permanent tissue damage e.g., osteoporosis, ophthalmopathy, myopathy, and personality changes. The diagnostic is based on the recognition of the symptoms of hyperthyroidism, goiter, eye signs, and the measurement of thyroid hormones and TSH.

Measurement of TRAbs is important to confirm GD and rule out other causes of hyperthyroidism making the diagnosis much more accurate [7–9] and can be critical in cases of Graves' Ophthalmopathy (GO) [10, 11] without hyperthyroidism. Since the discovery of TRAb more than 60 years ago as long-acting thyroid stimulator (LATS) by Adams and Purves [12], the measurement of TRAb has progressively improved in sensitivity, specificity, reliability, and usability and this has expanded its clinical application as reflected in many recent guidelines and surveys [13–17].

TRAb tests can be divided into main two categories depending on the detection method used: competition immunoassays and bioassays. Competition immunoassays detect all types of anti-TRAbs by measuring their ability to compete with a labeled ligand (TSH or a monoclonal antibody (MoAb) to TSH-R) for binding to the TSH receptor. Bioassays can detect the stimulating or blocking effect of the TRAb by measuring the production cAMP, the TSH-R intracellular signal, by cells expressing the TSH-R [11, 14]. Even if discrimination among the types of TRAbs could be of great interest in given clinical situations, such as in cases of unexplained changes in the thyroid function during or after pregnancy [18], competition immunoassays, that are easier, faster, and can be automated, are the tests commonly used in clinical diagnostic laboratories.

TRAb are autoantibodies and as such, they are not a molecularly defined analyte but a mixture of high-affinity IgG that bind selected epitopes of the TSH-R that varies among individuals and fluctuates within one individual. Small changes in the level, affinity, or fine specificity of the TRAb can result in major changes in their capacity to activate the TSH-R. Measuring TRAbs is, therefore, challenging, and generations of tests using different TSH-R preparations and ligands have been developed over the years, while in parallel, labeling and detection methods have also improved for the immunoassays in general. Therefore, the number of reports comparing tests in terms of sensitivity, specificity, safety, and cost-effectiveness as applied to different populations, have grown over the years [19–25] reflecting that none of them have yet met all the expectations of the clinical endocrinologist.

Therefore, the objective of this work is to compare the accuracy in terms of sensitivity and specificity of two automated 3$^{rd}$ generation immunoassays in use for TRAb testing of a clinical laboratory.

## Materials and methods

A comparative study was designed and conducted from February 2019 to July 2019 and the STARD guidelines were followed. The project was carried out at the Clinical Laboratories of Hospital Universitario Dr. Peset in Valencia (Spain), ISO 9001:2015 certified.

Informed consent was not requested as per ethics committee indication. The reason is that data was appropriately anonymized and all determinations were requested by specialists as part of the routine diagnostic testing. The study just included the determination requested by the specialists performed with two different methods. Medical decisions were taken according to the Hopitals' routine testing method results.

**Table 1. Demographic and clinical data.**

|  | NDGD | GD-R | GD-T | AIT | MNG | OTD | MISC | TOTAL |
|---|---|---|---|---|---|---|---|---|
| **Number** | 29 | 27 | 76 | 18 | 35 | 56 | 60 | 301 |
| **Female (%)** | 86.2 | 92.6 | 71.4 | 83.3 | 88.6 | 71.4 | 73.3 | 80.1 |
| **Age years (mean ± SD)** | 49.4 ±18.5 | 48.8 ±14.8 | 48.1 ±15.1 | 43.2 ±13.7 | 62.9 ±14.0 | 52.4 ±17.2 | 50 ±19.5 | 50.9 ±17.1 |

Abbreviations: GD, Graves' Disease GD-R, in remission; GD-T, under treatment; GDGO, GD Orbitopathy; NDGD, newly diagnosed; AIT Autoimmune Thyroiditis; MNG multinodular goiter; OTD, other thyroid diseases; Misc, Miscellaneous patients

## Patients

Baseline demographics and clinical characteristics of patients are given in Table 1.

Participants were divided in different clinical groups according to their diagnosis (Table 1 and Fig 1):

Group 1: 29 newly diagnosed untreated GD patients (NDGD)

Group 2: composed by 27 cases of GD patients in remission (GD-R)

Group 3: composed by 76 cases of GD patients under treatment, GD-T

Group 4: 18 cases of Autoimmune Thyroiditis (AIT), composed by 11 classical Hashimoto's thyroiditis and 7 postpartum thyroiditis, as assessed by TPO positive antibodies, imaging and hormones levels.

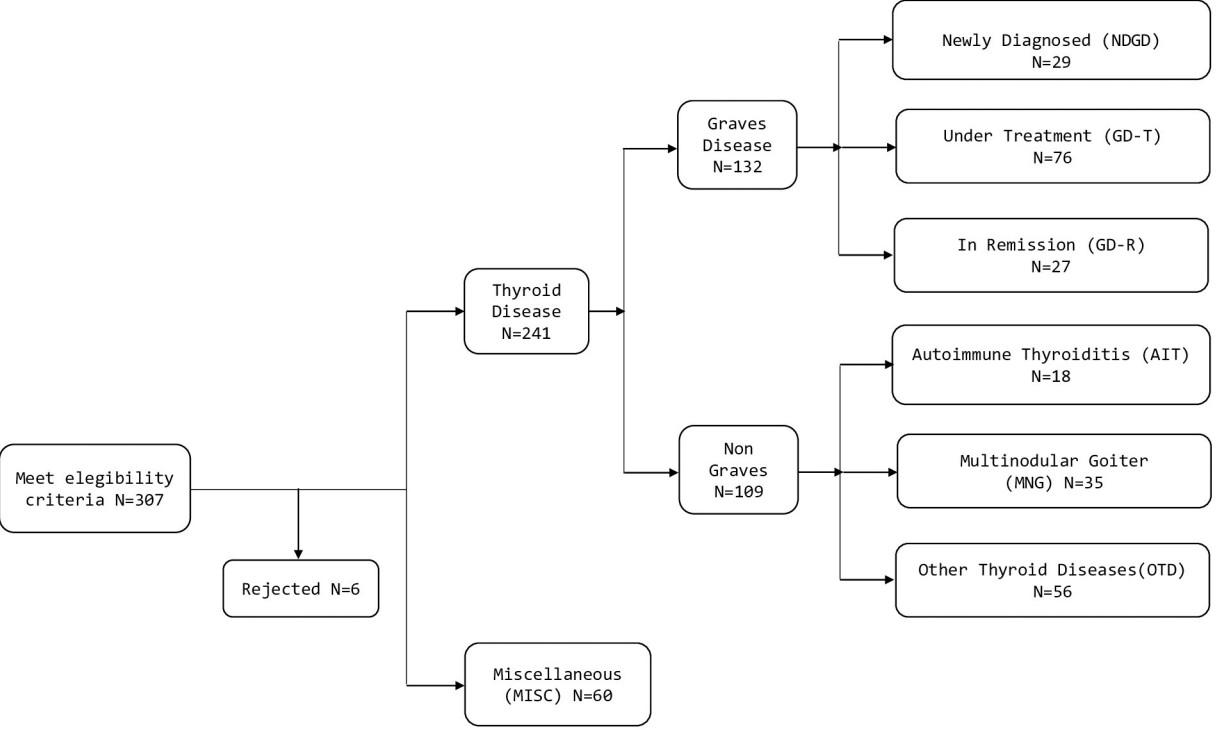

**Fig 1. Patient distribution in different clinical groups according to the final diagnosis.**

Group 5: 35 patients with non-toxic Multinodular Goiter (MNG) as assessed by history, image, and hormone levels.

Group 6: Other Thyroid Diseases (OTD), 56 patients with thyroid diseases other than GD, i.e., non-immune hyper or hypothyroidism, iatrogenic thyroid disorders, single thyroid nodules, and thyroid cancer.

Group 7: Miscellaneous (MISC), 60 cases referred to our laboratory because of diseases unrelated to thyroid pathology. This group was considered the control group in the study.

We did not make a category of patients with Graves' Orbitopathy (GO), due to the low number of patients with GO (n = 3).

Thyroid pathology diagnoses were established by qualified endocrinologists from Endocrinology and Nutrition Service from Hospital Universitario Dr. Peset, according to the American Thyroid Association and European Thyroid Association guidelines [13, 15].

## Samples

**Eligibility criteria.**   Consecutive samples received at the laboratory for TRAb testing from both Primary Care and Endocrinology specialists. To be included samples had to conform to the general pre-analytical requirements of the laboratory as established in the corresponding Standard Operating Procedure. None was excluded because of hemolysis, lipemia, or bilirubin, as these features do not interfere with the tests according to the specifications of the method. A total of 307 samples were initially tested but 6 were excluded from analysis, 2 due to incomplete clinical information, and 4 because they were duplicates. Results from 301 samples were used for the statistical analysis.

## TRAb test measurement methods

TRAb were measured in freshly obtained or -80ºC kept serum samples in parallel by two immunoassays, the fluorescence enzyme immunoassay (FEIA) EliA™ anti-TSH-R that uses the Phadia 250 auto-analyzer (Phadia A.B, Thermo Fisher Scientific, Uppsala, Sweden) and the electrochemiluminiscent immunoassay (ECLIA) Elecsys® anti-TSH-R method run in a COBAS 6000 auto-analyzer (Roche Diagnostics International Ltd, Basel, Switzerland).

The EliA™ anti-TSH-R test is an immunoassay that detects both stimulating and blocking antibodies. It is a competitive assay based on a human recombinant TSH-R (rTSH-R) immobilized to a solid phase (wells). Serum samples are dispensed to the wells for TRAb autoantibodies to bind to the TSH-R. A β-galactosidase labeled mouse monoclonal stimulating anti-TSH-R antibody is added in a second step. This MoAb binds the TSH-R epitopes that remain free after the patients' antibodies binding. After washing the free antibodies, the substrate for generating the fluorochrome is added and after a short incubation, the reaction is arrested and fluorescence measured: intensity is inversely proportional to TRAb in the problem sample. Finally, the fluorescence readings are converted into a titer of anti-TSH-R antibodies using a 6-point calibration curve [24].

The Elecsys® anti-TSH-R method (Roche Diagnostics Ltd, Basel, Switzerland), is also an immunoassay that works using the same competitive principle as EliA™ test, but with three major differences: it uses an immobilized purified porcine TSH-R (pTSH-R) as antigen, a human monoclonal stimulating autoantibody, M22, labeled with ruthenium as a competitor, and a chemiluminescent based detection. The more antibodies present in the patient serum sample, the less luminescence is detected in the final measurement [25].

Both assays were carried out according to manufacturer specifications. Their most relevant technical parameters are showed in Table 2.

**Table 2. Manufacturer specifications of the compared assays.**

|  | Elecsys | EliA™ |
|---|---|---|
| **Negative** | ≤1.75 IU/L | <2.9 IU/L |
| **Grey zone** | None | 2.9–3.3 IU/L |
| **Positive** | >1.75 IU/L | >3.3 IU/L |
| **LoQ** | 0.8 IU/L | 1.5 IU/L |
| **UpQ** | 40 IU/L | 68 IU/L |
| **Calibration standard** | 90/672 NIBSC | 08/204 NIBSC |
| **Assay time** | Depending on the system used | Depending on the system used |
| **Sample volume** | 45 μL | 50 μL |
| **Functional sensitivity** | 1.5 IU/L | 0.3 IU/L |
| **Precision (CV)** | Intra-run: 6.1% Inter-run: 6.13% | Intra-run: 4.7% Inter-run: 6.31% |

Abbreviations: LoQ = Lower limit of Quantitation. UpQ = Upper limit of Quantitation; NIBSC, National Institute Biologicals Standards and Controls (UK)

## Statistical analysis

The number of patients required for this study was estimated by extrapolation from reference reports describing similar accuracy comparative studies for TRAb measurement (24,25).

For calculating the diagnostic accuracy of the tests, results from patients already under treatment were not included as anti-thyroid drugs may reduce TRAb levels. Results in the grey zone for EliA™ test were considered negative in this comparison.

Statistical analysis was performed using SAS system 9.4 (SAS, Cary, USA) and GraphPad Prism (GraphPad Software, San Diego, USA) software packages. A two-sided value of $p < 0.05$ was considered statistically significant.

The results were compared as quantitative and qualitative variables, by applying Spearman's correlation and Passing-Bablok respectively. TRAb values below the lower limits and over the upper limits of quantitation (LoQ and UpQ respectively) were considered equal to limit values for statistical purposes. Bland-Altman test was not applicable as the Shapiro-Wilk test showed that TRAb results were not normally distributed. Results are given as median with range and the subsequent statistical analysis was performed with Cohen's kappa to study the agreement of categorical variables (positive vs. negative) and Kruskal-Wallis test to compare the groups of patients.

Receiver operating characteristic (ROC) curves were plotted and analyzed using Analyse-it Version 4.60 for Microsoft Excel (Leeds, UK) to compare the area under the curve (AUC) by both methods. Clinical sensitivity, specificity, predictive value, and likelihood ratio (LR) were also calculated.

## Results and discussion

### Anti-TSH-R immunoassays performance comparison

The main results are summarized in Table 3. Spearman correlation showed significant positive linearity of results for both tests (0.725). It was therefore adequate to apply the Passing-Bablok test where the titers from both tests were shown not to be comparable as they showed systematic and proportional differences (Fig 2).

Bland-Altman plot was performed for a specific descriptive purpose, as the result differences were not normally distributed according to Shapiro-Wilk test p<0.0001 (Fig 3) and therefore this is not the most adequate representation for the results of these tests.

**Table 3. Statistical analysis comparing results from Elecsys® and EliA™ assays.**

| TEST | | RESULT |
|---|---|---|
| **Spearman correlation** | | rho = 0.725 p < 0.0001 |
| **Passing-Bablock** | | A = -1.6891 (-1.9436 to -1.4675) |
| | | B = 1.2732 (1.1750 to 1.3786) |
| **Shaphiro-Wilk** | | p < 0.0001 |
| **Cohen's Kappa** | | K = 0.8176 (0.7486 to 0.8866) |
| **Kruskal-Wallis** | **Elecsys®** | NDGD vs GD, AIT, MNG, OTD, MISC p < 0.001 |
| | **EliA™** | NDGD vs GD, AIT, MNG, OTD, MISC p < 0.001 |
| **ROC** | **Elecsys®** | AUC 0.995 |
| | | Cut-off 1.75 IU/L |
| | | Sensitivity 100% |
| | | Specificity 95.3% |
| | | LR+ 21.1 |
| | **EliA™** | AUC 0.996 |
| | | Cut-off 3.3 IU/L |
| | | Sensitivity 96.6% |
| | | Specificity 99.4% |
| | | LR+ 163.2 |

Abbreviations: AUC, area under de curve; GD, Graves' Disease GD-R, in remission; GD-T, under treatment; GDGO, GD Orbitopathy; NDGD, newly diagnosed; AIT Autoimmune Thyroiditis; MNG multinodular goiter; OTD, other thyroid diseases; MISC, Miscellaneous patients

Cohen's kappa index for categorized values was k = 0.8176, indicating strong agreement which was higher for negative than for positive results (99.5% and 78.5%, respectively).

The comparison between paired clinical groups, performed by Kruskal-Wallis test (Fig 4) showed that NDGD is statistically different from all the other groups for both Elecsys® and EliA™ methods, with the logical exception of GD-T.

NDGD, newly diagnosed untreated GD patients; GD-R, GD patients in remission; GD-T, GD patients under treatment; AIT, Autoimmune Thyroiditis; MNG, Multinodular Goiter non-toxic; OTD, Other Thyroid Diseases; MISC, Miscellaneous.

According to ROC analysis (Fig 5), the optimal cut-off (OCO) for EliA™ method is 3.2 IU/L, at which the EliA™ method has a sensitivity of 96.6% and a specificity of 99.4%

The sensitivity and specificity of Elecsys® method at the manufacturer given cut-off value, 1.75 IU/L, are 100% and 95.3% respectively.

The area under the curve (AUC) for Elecsys® was 0.995 and 0.996 for EliA™. They showed the high diagnostic efficiency of both methods with a minimal non-statistically significant difference.

The value of TRAb measurement in GD diagnosis and treatment monitoring is widely accepted in clinical practice and it is now included in scientific society guidelines. Even though currently used competitive assays do not distinguish between the different types of TRAb activities, overall binding TSH inhibition measurement is sufficient for supporting clinical diagnosis in most cases. The exceptions are a few complex clinical situations such as neonatal hyperthyroidism where a bioassay could be useful [14].

Given the different calibration standards used, Elecsys® and EliA™ results are not directly comparable. This lack of direct comparability among 3rd generation methods has already been reported [21]. Articles comparing different TRAb assays including the EliA™ or the Elecsys®

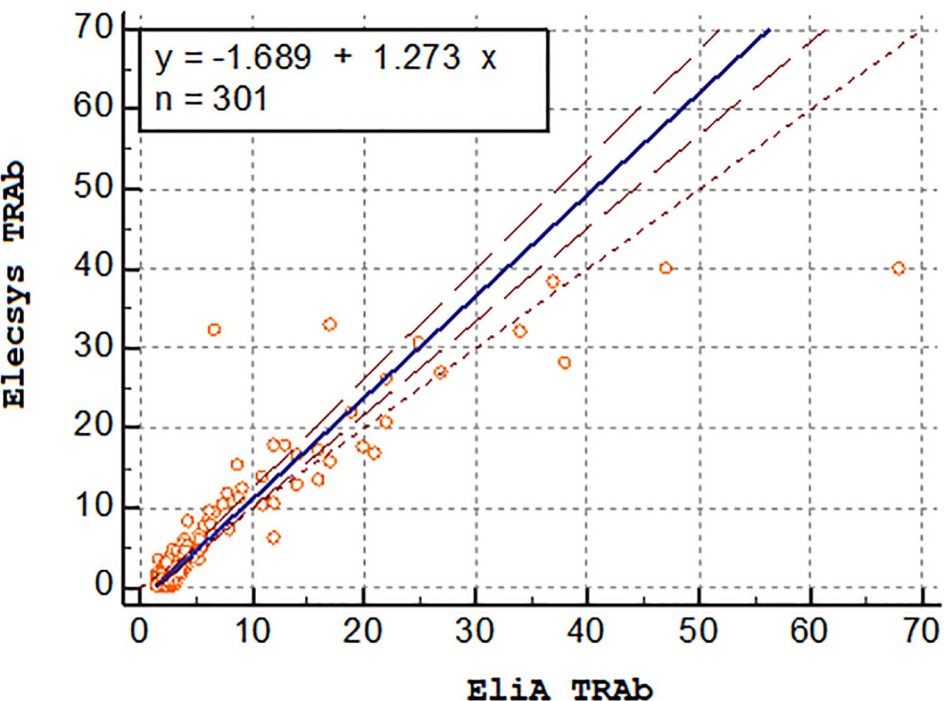

**Fig 2. Passing-Bablok regression.** Correlation between EliA™ and Elecsys® TRAb. TRAb are displayed in UI/l. Dotted line (- - - -) corresponds to identity. Blue line corresponds to Passing Bablock regression and double dotted line (- - - - - -) to 95% CI bands. Tests values are displayed as orange circles.

tests have been published, but to our knowledge, this is the first head-to-head direct comparison (Table 4).

In our analysis, the Elecsys® shows slightly higher sensitivity while the EliA™ test a slightly higher specificity. Villalta et al. [24] compared EliA™ with TRAK™ RIA, Brahms (Thermo Scientific, Hennigsdorf, Germany) and Immulite™ TSI assay (Siemens Healthcare, Llanberis, UK) but not the Elecsys. For the EliA™ test the reported specificity of 99.6% is very similar to the 99.4% in our study, but in our series the optimal cut-off of 3.2 IU/L is closer to the upper limit of normality provided by the manufacturer, and slightly lower than the optimal cut-off of 3.8 IU/L reported in their paper.

In the report of Struja et al. [25], the EliA™ test was the most specific (97.9% at the manufacturer cut-off of 3.3 IU/L) and showed a specificity of 97.7% at the same cut-off in Luthers' work [26]. Sensitivity was much lower in Struja and Luther series (71.7% and 79% respectively), but both studies included GD patients under treatment while we did not.

In the paper of Villalta et al. [24] the EliA™ sensitivity for untreated GD was of 94.7% which is slightly lower than our result of 96.6%. Adopting their cut-off point to our series, the sensitivity would be down to 73.8% because some of our untreated GD patients (5/29) are in the low positive range, from 3.3 to 3.8 IU/L.

In relation to Elecsys® anti-TSH-R method, Dourudian et al. found a sensitivity of 95% and a specificity of 97% when comparing their GD patients to a healthy control group [23] However, when they compared GD patients to a mixed group of healthy and disease controls, the specificity decreased to 90%. The sensitivity of Elecsys® in our series is 100%, as we did include any case of seronegative GD during the period of the study. We may have to re-assess this issue in future work.

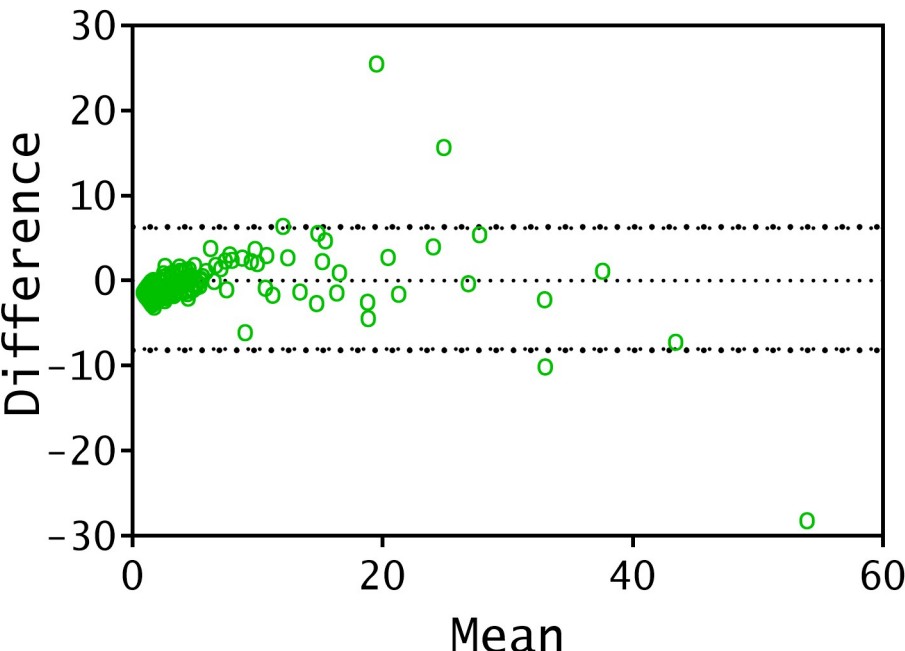

**Fig 3. Bland-Altman plot.** Inter-assay agreement between EliA™ and Elecsys® TRAb. The dashed line (........) corresponds to bias. Lower (-8.036) and upper (6.195) 95% limits of agreement are plotted as speckled lines. Y-axis plots the difference scores of EliA™ and Elecsys® TRAb measurements against the mean for each studied sample.

There is a good concordance between Elecsys® and EliA™ for negative results and a total concordance for clearly positive results. The controversy is focused on Elecsys® low positives that could be equivocal/negative for EliA™. But this is an expected result when comparing one assay that prioritizes sensitivity with another prioritizing specificity. In their comparative

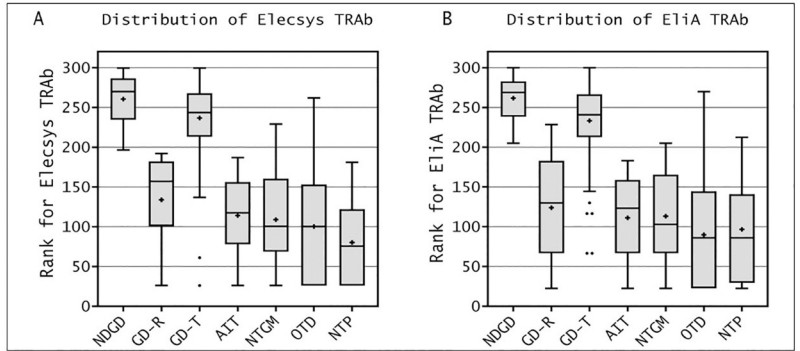

**Fig 4. Box-plot of Kruskal-Wallis analysis. A.** Elecsys® TRAb levels in the different groups enrolled in the study. **B.** EliA™ TRAb levels in the different groups enrolled in the study. TRAb levels are displayed as data ranks obtained by transforming to continue data ranks using Roche's test. For each box plot, the central line represents the median; the boxes limits represent the upper and lower quartile; black points are outliers (a value more than 1.5 times the interquartile range above/below the interquartile values); and the black lines are the whiskers, which extend from the interquartile ranges to the maximum values that are not classed as outliers.

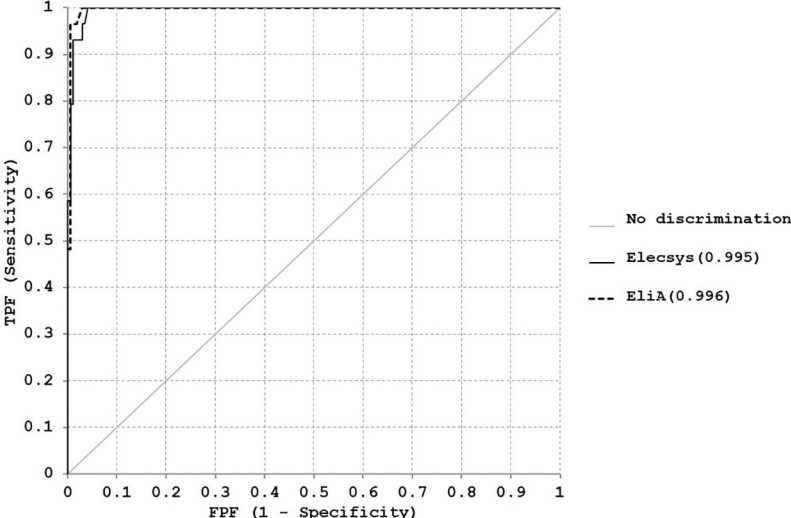

**Fig 5. ROC (Receiver operating curve) analysis for EliA™ and Elecsys®.** ROC curve of both methods at manufacturer cut-off values.

study, Autilio et al. [27] found the four near cut-off false positives Elecsys® corresponding to chronic autoimmune thyroiditis cases. We also found six near cut-off positives by Elecsys® that by EliA™ were five negative and one equivocal, all of them corresponded to the OTD group.

A patient from OTD group with a nodular thyroid gland having a single dominant nodule was positive for both methods, but in this case, we cannot strictly consider this a false positive as overlapping cases of GD and multinodular thyroid are not rare.

The overall impression considering quantitative and qualitative results is that both Elecsys® and EliA™ are good methods for TRAb measurement in clinical laboratories. Whether to prioritize sensitivity or specificity and how to manage the results around cut-off and how to deal with grey zone results are difficult questions. The fundamentals of the two methods are different for these aspects. EliA™ prioritizes specificity and defines a grey zone to report intermediate results. Elecsys®, at the manufacturer's cut-off value of 1.75 IU/L, clearly prioritizes sensitivity.

In this study, we have focused on test performance, however, other test characteristics like assay time, sample volume, and precision are also worthy of evaluation when selecting the

**Table 4. Test performance values obtained from comparative studies for TRAb measurement methods including EliA™ and/or Elecsys® evaluation.**

| Study | Methods | Cut-off | Sensitivity | Specificity |
|---|---|---|---|---|
| **Current study (2019)** | EliA™ T-Fisher | 3.2 IU/L** | 96.6% | 99.4%†† |
| | Elecsys® Roche | 1.75 IU/L* | 100% | 95.3%†† |
| **Struja et al. (2019)** | EliA™ T-Fisher | 3.3 IU/L* | 71.1% | 97.9%†† |
| **Villalta et al. (2018)** | EliA™ T-Fisher | 3.8 IU/L** | 94.7% | 99.6%† |
| **Doroudian et al. (2017)** | Elecsys® Roche | 1.75 IU/L* | 95% | 97% † (90%†††) |
| **Luther et al. (2017)** | EliA™ T-Fisher | 3.3 IU/L* | 79% | 97.7%†† |

*Manufacturer suggested cut-off.

**Optimal cut-off according to ROC.

†Specificity value comparing GD to Healthy Controls.

††Specificity value Comparing GD to Disease Controls.

†††Specificity value comparing GD to Healthy and Disease Controls (MTNG/Primary Autoimmune Hyperthyroidism).

appropriate test for a specific laboratory. Nevertheless, it is difficult to draw general conclusions, as their impact mainly depends on the type of laboratory system used, and the laboratory workflow. Sample volume is not critical in daily routine, however, lower volume sample needs are always better for patients.

As other studies have suggested [28], our results support the use of a higher cut-off point for the Elecsys® test in clinical laboratory use, as our preference is to prioritize a higher specificity. In principle, for low prevalence diseases as most autoimmune diseases, a small reduction in specificity is translated into an excess of false positives, as compared to false negatives caused by a small reduction in sensitivity. Moreover, this test is much more a confirmatory test than a screening test in the diagnostic strategy of GD, addressing a parameter in which positivity has been sometimes considered pathognomonic [29, 30].

In the same line with the results from Villalta et al. [24] series for Hashimoto's thyroiditis patients, we observed an absence of TRAb positivity in our AIT patients group.

In relation to establishing a single cut-off or considering grey zones, both ways to report results are perfectly valid if the interpretation is carefully reported to the clinicians. It is also important that clinicians understand the limitations of the technique in use, and this is the role of clinical laboratory specialists.

Our study has two main limitations, first, we did not include a healthy patients group as the negative control, which is always good to have. Second, we did not evaluate the performance of both tests for patients' disease follow-up and relapse prediction. This is also a key point when deciding which test to include in the daily routine, and our next objective as follow-up performance data is scarce on TRAb measurement methods.

## Conclusions

In conclusion, both methods are perfectly valid and useful for routine TRAb analysis, with a trend to specificity in the case of EliA™. Moreover, it is very important to raise awareness to the clinicians on the proper interpretation of results falling in the grey zone or close to cut-off values. Given the increasing agreement on literature about TRAb utility in GD diagnosis and management and in order to minimize costs, TRAb measurements must be appropriately integrated into the routine of GD diagnosis and management.

## Supporting information

**S1 Graphical abstract.**
(TIF)

## Acknowledgments

We greatly acknowledge to Dr. Ricardo Pujol Borrell -Immunology Division Senior Consultant at Hospital Universitari Vall D'Hebrón- for his kind support during manuscript edition and Dr. Elisa Caballero Calabuig and Dr. Marcelino Gómez Balaguer—Nuclear Medicine and Endocrinology Services from Hospital Universitario Dr. Peset- for their kind advice and collaboration to clinical data evaluation. Finally, we want to acknowledge Thermo Fisher Scientific for providing TSH-R tests.

## Author Contributions

**Conceptualization:** José María López Ortega, Pilar Salvador Martínez.

**Data curation:** José María López Ortega.

**Formal analysis:** José María López Ortega.

**Funding acquisition:** José María López Ortega.

**Investigation:** José María López Ortega, Delia Acevedo-León, Núria Estañ Capell.

**Methodology:** José María López Ortega.

**Project administration:** José María López Ortega.

**Resources:** José María López Ortega.

**Supervision:** José María López Ortega, Pilar Salvador Martínez, Delia Acevedo-León, Núria Estañ Capell.

**Validation:** José María López Ortega, Pilar Salvador Martínez, Delia Acevedo-León, Núria Estañ Capell.

**Writing – original draft:** José María López Ortega.

**Writing – review & editing:** José María López Ortega, Pilar Salvador Martínez, Delia Acevedo-León, Núria Estañ Capell.

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
