## [Decision Letter · Decision Letter 0]

8 Mar 2022

PONE-D-21-37268Anti-TSH receptor antibodies (TRAb): comparison of two third generation automated immunoassays broadly used in clinical laboratories and results interpretationPLOS ONE

Dear Dr. Jose Maria Lopez Ortega,

Thank you for submitting your manuscript to PLOS ONE. After careful consideration, we feel that it has merit but does not fully meet PLOS ONE’s publication criteria as it currently stands. Therefore, we invite you to submit a revised version of the manuscript that addresses the points raised during the review process.

We look forward to receiving your revised manuscript.

Kind regards,

Purvi Purohit

Academic Editor

PLOS ONE

Journal Requirements:

Additional Editor Comments (if provided):

The manuscript needs major revisions and is not acceptable in its current format. If the authors wish to continue they should address the queries raised by the reviewer.

Reviewers' comments:

Reviewer's Responses to Questions

**Comments to the Author**

1. Is the manuscript technically sound, and do the data support the conclusions?

Reviewer #1: Partly

2. Has the statistical analysis been performed appropriately and rigorously? 

Reviewer #1: Yes

3. Have the authors made all data underlying the findings in their manuscript fully available?

Reviewer #1: Yes

4. Is the manuscript presented in an intelligible fashion and written in standard English?

Reviewer #1: Yes

5. Review Comments to the Author

Reviewer #1: José María López Ortega et al. manuscript "Anti-TSH receptor antibodies (TRAb): comparison of two third generation automated immunoassays broadly used in clinical laboratories and results interpretation" studied the comparison of accuracy in terms of sensitivity and specificity of two automated IIIrd generation immunoassays in use for TRAb testing of a clinical laboratory, the recent EliATM method (ThermoFisher Scientific, Germany) and the Elecsys® /Cobas ECLIA (Roche Diagnostics Ltd, Switzerland). This paper is easy to understand and read. Some of the aspects to be improved to favor its understanding is the methodology and result sections. I would like to add some comments and suggestions to this paper.

1. Please write a full form of abbreviation (TRAb) {Page No.4; Line No. 67}

2. I think the cost per sample matters a lot here, it is very important to compare it, today most of the third world countries are facing the lowest level of health services, so the lowest cost of any diagnostics test is to reach every poor patient.

3. Apart from this, by doing a comparative analysis on the following points, their explanation can be helpful in improving this Manuscript.

(A) Assay Time

(B) sample volume

(C) Measuring Range

(D) Functional Sensitivity

(E) Cutoff

(F) Precision

4. Please correct typing error. {Page No.15; Line No. 277}

"For the EliATM test the reported specificity of 99.6% is very similar to the 99.4% in our study, but in our series the optimal cut-off of 3,2 IU/L is closer the upper limit of normality provided by the manufacturer, and slightly lower than the optimal cut-off of 3.8 IU/L reported in their paper."

5. Please explain "disease control" in the manuscript.

6. In Figure 3, denote each specific sample in relation to disease.

7. Many figure captions do not explain what colouring and boxing mean in many figures. The figure legend needs a lot of improvement.

8. I would like to see a critical discussion on limitations in the current study.

9. The authors perform very similar analyses to https://doi.org/10.1186/s12902-019-0363-6, except I would argue the linked paper is more thorough since they use more rigorous statistical analysis. The authors should justify what their paper adds to the field that this paper doesn’t.

6. PLOS authors have the option to publish the peer review history of their article (what does this mean?). If published, this will include your full peer review and any attached files.

Reviewer #1: No

---

## [Author Response · Author response to Decision Letter 0]

7 Jun 2022

Dear reviewers, 

Thank you very much for the time you've dedicated to evaluate our work. Please find below the response to all the comments:

2. Reviewer 1

Comment from the author: all changes applied to the manuscript due to reviewer suggestions are highlighted in the manuscript in light blue.

2.1. Comment 1

Please write a full form of abbreviation (TRAb) {Page No.4; Line No. 67}

Response

The full form abbreviation has been written. See below:

“functional effect on TSH-R, three types of TRAb or Thyrotropin Receptor Autoantibodies (TRAb) can be considered:”

2.2. Comment 2

I think the cost per sample matters a lot here, it is very important to compare it, today most of the third world countries are facing the lowest level of health services, so the lowest cost of any diagnostics test is to reach every poor patient.

Response

We completely agree with the reviewer comment, but we were not able to conduct a health economic evaluation. It must be taken into account that, in our country, health assistance is free for all citizens, so we are focusing on offering solutions that provide better aid in the diagnosis of patients. Moreover, the price of a test is subjected to many variables (public tenders, test volume, etc.). 

A reader from third world countries must evaluate the results of this study in the context of the country-specific prices for the methods, and select the best option for them.

2.3. Comment 3

3. Apart from this, by doing a comparative analysis on the following points, their explanation can be helpful in improving this Manuscript.

(A) Assay Time

(B) sample volume

(C) Measuring Range

(D) Functional Sensitivity

(E) Cut-off

(F) Precision

Response

Again we agree with reviewer comments. Some of the aforementioned assay characteristics were already described in table 2 (measuring range, functional sensitivity, and cut-off) but we have added the ones missing (assay time, sample volume, and precision). Also, we have added the following paragraph to the discussion (page 16, line 310)

“In this study, we have focused on test performance, however, other test characteristics like assay time, sample volume, and precision are also worthy of evaluation when selecting the appropriate test for a specific laboratory. Nevertheless, it is difficult to draw general conclusions, as their impact mainly depends on the type of laboratory system used, and the laboratory workflow. Sample volume is not critical in daily routine, however, lower volume sample needs are always better for patients.”

2.4. Comment 4

Please correct typing error. {Page No.15; Line No. 277}. 

Response

The typing error has been corrected. See below:

“99.4% in our study, but in our series the optimal cut-off of 3.2 IU/L is closer the”

2.5. Comment 5

Please explain "disease control" in the manuscript.

Response

We did not include a “disease control group” if it is defined as “healthy population”. However, we included 60 samples received for TRAb measurement corresponding to diagnostics unrelated to thyroid pathologies. With the exception of a couple of patients with Diabetes Mellitus type I, samples at this group corresponded to patients under control and not affected by autoimmune diseases or under clinical processes of a minor entity. We have modified the text accordingly to make it clear (pag 7, line 131):

“Group 7: Miscellaneous (MISC), 60 cases referred to our laboratory because of diseases unrelated to thyroid pathology. This group was considered the control group in the study.”

2.6. Comment 6

In Figure 3, denote each specific sample in relation to disease.

Response

We would like to denote each specific sample in relation to disease as the reviewer comments, however a modification like the proposed requires the involvement of the statistician and, due to the short timespan we’ve had, It has not been possible. As it is a clear improvement, but we do not consider it critical, we leave it to your criteria to include the modification for the final publication if you think it is essential, and of course, the manuscript is accepted.

2.7. Comment 7

Many figure captions do not explain what coloring and boxing mean in many figures. The figure legend needs a lot of improvement.

Response

We completely agree with reviewer's comments. We have modified figure captions in order to better explain them. See below the modified figure captions:

“Fig 1. Patient distribution in different clinical groups according to the final diagnosis. 

Fig 2. Passing-Bablok regression. Correlation between EliA™ and Elecsys® TRAb. Trab are displayed in UI/l. Dotted line (- - - -) corresponds to identity. Blue line corresponds to Passing Bablock regression and double dotted line (-- – – –) to 95% CI bands. Tests values are displayed as orange circles.

Fig 3. Bland-Altman plot. Inter-assay agreement between ELIA™ and Elecsys® TRAb. The dashed line (........) corresponds to bias. Lower (-8.036) and upper (6.195) 95% limits of agreement are plotted as speckled lines. Y-axis plots the difference scores of ELIA™ and Elecsys® TRAb measurements against the mean for each studied sample.

Fig 4. Box-plot of Kruskal-Wallis analysis. A. Elecsys®TRAb levels in the different groups enrolled in the study. B. EliATM TRAb levels in the different groups enrolled in the study. TRAb levels are displayed as data ranks obtained by transforming to continue data ranks using Roche’s test. For each box plot, the central line represents the median; the boxes limits represent the upper and lower quartile; black points are outliers (a value more than 1.5 times the interquartile range above/below the interquartile values); and the black lines are the whiskers, which extend from the interquartile ranges to the maximum values that are not classed as outliers.

NDGD, newly diagnosed untreated GD patients; GD-R, GD patients in remission; GD-T, GD patients under treatment; AIT, Autoimmune Thyroiditis; MNG, Multinodular Goiter non-toxic; OTD, Other Thyroid Diseases; Misc, Miscellaneous.

Fig 5. ROC (Receiver operating curve) analysis for EliATM and Elecsys®. ROC curve evaluates the accuracy of both methods at manufacturer cut-off values, showing the tend to the specificity of EliATM assay.”

2.8. Comment 8

I would like to see a critical discussion on limitations in the current study.

Response

We especially agree with the reviewer. All manuscripts must include a description of the limitations of the study in the discussion section and our manuscript does not include it. Thus, have included the following paragraph in the discussion (page 13, line 248):

“Our study has two main limitations, first, we did not include a healthy patients group as the negative control, which is always good to have. Second, we did not evaluate the performance of both tests for patients' disease follow-up and relapse prediction. This is also a key point when deciding which test to include in the daily routine, and our next objective as follow-up performance data is scarce on TRAb measurement methods.”

2.9. Comment 9

The authors perform very similar analyses to https://doi.org/10.1186/s12902-019-0363-6, except I would argue the linked paper is more thorough since they use more rigorous statistical analysis. The authors should justify what their paper adds to the field that this paper doesn’t.

Response

Struja’s manuscript is really interesting for groups working in the field. However, although Struja’s comparative study includes EliATM anti-TSH-R, it does not include Elecsys® anti-TSH-R. EliATM and Elecsys® are two of the main used methods in our environment and that's why we decided to perform this specific comparison, to have a head-to-head comparison between two methods that have never been compared before, and that can be useful for professionals working in the field.

In fact, we highlight the novelty of the comparison in the manuscript (pag 13, line 261). 

“Articles comparing different TRAb assays including the EliATM or the Elecsys® tests have been published, but to our knowledge, this is the first head-to-head direct comparison (Table 4).”

---

## [Editor Report · Decision Letter 1]

21 Jun 2022

Anti-TSH receptor antibodies (TRAb): comparison of two third generation automated immunoassays broadly used in clinical laboratories and results interpretation

PONE-D-21-37268R1

Dear Dr. José María López Ortega

We’re pleased to inform you that your manuscript has been judged scientifically suitable for publication and will be formally accepted for publication once it meets all outstanding technical requirements.

Kind regards,

Purvi Purohit

Academic Editor

PLOS ONE

Additional Editor Comments (optional):

Congratulations to the authors. The manuscript has been revised satisfactorily and can now be accepted for publication.

---

## [Editor Report · Acceptance letter]

14 Jul 2022

PONE-D-21-37268R1 

Anti-TSH receptor antibodies (TRAb): comparison of two third generation automated immunoassays broadly used in clinical laboratories and results interpretation 

Dear Dr. López Ortega:

I'm pleased to inform you that your manuscript has been deemed suitable for publication in PLOS ONE. Congratulations! Your manuscript is now with our production department. 

Kind regards, 

on behalf of

Dr. Purvi Purohit 

Academic Editor

PLOS ONE